# A Variational Approach for Learning from Positive and Unlabeled Data

**Hui Chen**[*]
School of Mathematical Science
Tongji University, Shanghai, P. R. China
`hui.chen96@outlook.com`

**Fangqing Liu**[*]
School of Mathematical Science
Tongji University, Shanghai, P. R. China
`fangqingliu0@gmail.com`

**Yin Wang**
School of Electronics and Information Engineering
Tongji University, Shanghai, P. R. China
`yinw@tongji.edu.cn`

**Liyue Zhao**
Cloudwalk Inc.
Shanghai, P. R. China
`zhaoliyue@cloudwalk.cn`

**Hao Wu**[†]
School of Mathematical Science
Tongji University, Shanghai, P. R. China
`hwu@tongji.edu.cn`

## Abstract

Learning binary classifiers only from positive and unlabeled (PU) data is an important and challenging task in many real-world applications, including web text classification, disease gene identification and fraud detection, where negative samples are difficult to verify experimentally. Most recent PU learning methods are developed based on the misclassification risk of the supervised learning type, and they may suffer from inaccurate estimates of class prior probabilities. In this paper, we introduce a variational principle for PU learning that allows us to quantitatively evaluate the modeling error of the Bayesian classifier directly from given data. This leads to a loss function which can be efficiently calculated without involving class prior estimation or any other intermediate estimation problems, and the variational learning method can then be employed to optimize the classifier under general conditions. We illustrate the effectiveness of the proposed variational method on a number of benchmark examples.

## 1 Introduction

In many real-life applications, we are confronted with the task of building a binary classification model from a number of positive data and plenty of unlabeled data without extra information on the negative data. For example, it is common in disease gene identification [1] that only known disease genes and unknown genes are available, because the reliable non-disease genes are difficult to obtain. Similar scenarios occur in deceptive review detection [2], web data mining [3], inlier-based outlier detection [4], etc. Such a task is certainly beyond the scope of the standard supervised machine learning, and where *positive-unlabeled* (PU) learning comes in handy. A lot of heuristic approaches [5–8] were proposed by identifying reliable negative data from the unlabeled data, which heavily rely on the choice of the heuristic strategies and the assumption of data separability (i.e., positive

---

[*]Equal Contribution
[†]Corresponding Author

and negative data are non-overlapping). The rank pruning (RP) [9] provides a more general way by regarding the PU learning as a specific positive-negative learning problem with noisy labels , but the data separability is still necessary for the consistent noise estimation.

The risk estimator developed in [10, 11] promises an effective solution to PU learning. It calculates the risk of a classifier $\Phi$ by

$$\text{risk}(\Phi) = \pi_P \mathbb{E}_{\text{labeled data}} \left[ \ell_+ \left( \Phi(x) \right) - \ell_- \left( \Phi(x) \right) \right] + \mathbb{E}_{\text{unlabeled data}} \left[ \ell_- \left( \Phi(x) \right) \right] \quad (1)$$

and can achieve an unbiased estimation of the expected misclassification risk (in the sense of supervised learning) via empirical averaging, where $\ell_+, \ell_-$ denotes the misclassification loss on positive and negative data respectively, and $\pi_P = \mathbb{P}(y = +1)$ denotes the *class prior*, i.e., the proportion of positive data in the unlabeled data. Then the classifier can be trained through minimization of the estimated risk when $\pi_P$ is known. However, such a method easily leads to severe overfitting. In order to address this difficulty, a non-negative risk estimator is presented in [12], which is biased but more robust to statistical noise. Another type of misclassification risk based method, called PULD, was proposed in [13], where PU learning is formulated as a maximum margin classification problem for a given $\pi_P$, and can be solved by efficient convex optimizers. But this method is applicable only for linear classifiers in non-trainable feature spaces.

Recently, applications of generative adversarial networks (GAN) in PU learning also have received growing attention [14, 15], where the generative models learn to generate fake positive and negative samples (or only negative samples), and the classifier is trained by using the fake samples. Experiments show that GAN can improve the performance of PU learning when the size of positive labeled data is extremely small, and the asymptotic correctness can be proved under the condition that the exact value of $\pi_P$ is available [14].

**Problems of class prior estimation**    The class prior $\pi_P$ plays an important role in PU learning as analyzed previously, but it cannot be automatically selected as a trainable parameter. As an example, when trying to minimize the risk defined in (1) w.r.t. both $\pi_P$ and the classifier, we obtain a trivial solution with $\pi_P = 1$ and all data being predicted as positive ones. Furthermore, it is also difficult to adjust $\pi_P$ as a hyper-parameter by cross validation unless some negative data are available in the validation set. Hence, in many practical applications, class prior estimation methods [16–19] are required, which usually involve kernel machines and are quite computationally costly. Moreover, the experimental analysis in [12] shows that the classification performance could be badly affected by an inaccurate estimate.

**Contributions**    In view of the above remark, it is natural to ask if *an accurate classifier can be obtained in PU learning without solving the hard class prior estimation problem as an intermediate step*. Motivated by this question, we introduce in this paper a variational principle for PU learning, which allows us to evaluate the difference between a given classifier and the ideal Bayesian classifier in a class prior-free manner by using only distributions of labeled and unlabeled data. As a consequence, one can efficiently and consistently approximate Bayesian classifiers via variational optimization. Our theoretical and experimental analysis demonstrates that, in contrast with the existing methods, the variational principle based method can achieve high classification accuracies in PU learning tasks without the estimation of class prior or the assumption of data separability. A brief algorithmic and theoretical comparison of VPU and selected previous schemes is provided in Table 1.

## 2    Problem setting and notations

Let us consider a binary classification problem where features $x \in \mathbb{R}^d$ and class labels $y \in \{-1, +1\}$ of instances are distributed according to a joint distribution $\mathbb{P}(x, y)$. Suppose that we have a positive dataset $\mathcal{P} = \{x_1, \dots, x_M\}$ and an unlabeled dataset $\mathcal{U} = \{x_{M+1}, \dots, x_{M+N}\}$. The goal of PU learning is to find a binary classifier based on $\mathcal{P}$ and $\mathcal{U}$ so that class labels of unseen instances can be accurately predicted. In this work, we aim to approximate the ideal Bayesian classifier $\Phi^*(x) \triangleq \mathbb{P}(y = +1|x)$ with a parametric model $\Phi$ based on the following assumptions:

**Assumption 1.** *Labeled and unlabeled data are independently drawn as*

$$\mathcal{P} = \{x_i\}_{i=1}^M \overset{\text{i.i.d}}{\sim} f_P, \quad \mathcal{U} = \{x_i\}_{i=M+1}^{M+N} \overset{\text{i.i.d}}{\sim} f \quad (2)$$

Table 1: A comparison of PU learning methods. Here uPU and nnPU are proposed in [10, 12], GenPU is presented in [14], rank pruning [9] is developed within the framework for classification with noisy labels, the Rocchio-SVM method proposed in [20] is a representative method developed based on identification of reliable negative data, and PULD [13] is proposed based on the large margin strategy. Rank pruning can be implemented with unknown class prior, but it contains an estimator for class prior explicitly and the estimator is consistent only in the case of data separability.

| Method | Training without class prior $\mathbb{P}(y = +1)$ or its estimate | Consistency or optimality without assumption of data separability |
|--------|:---:|:---:|
| VPU | ✓ | ✓ |
| uPU/nnPU | × | ✓ |
| GenPU | × | × |
| Rank pruning | ✓ | × |
| Rocchio-SVM | ✓ | × |
| PULD | × | ✓ |

where $f_P \triangleq \mathbb{P}(x|y = +1)$ is the distribution of the positive class and $f(x) \triangleq \mathbb{P}(x)$ denotes the marginal distribution of the instance feature.

**Assumption 2.** There exists a set $\mathcal{A} \subset \mathbb{R}^d$ satisfying $\int_{\mathcal{A}} f_P(x)\mathrm{d}x > 0$ and

$$\Phi^*(x) = 1, \quad \forall x \in \mathcal{A}. \tag{3}$$

Here, Assumption 1 is the traditional *selected completely at random* (SCAR) assumption in PU learning [21, 11]. Assumption 2 implies that a set of $x$ are almost surely positive, which is approximately satisfied in most practical cases and actually a strong variant of the *irreducibility* assumption in literature of mixture proportion estimation of PU data [22] (see Section A.2 in Suppl. Material). In practice, $\mathcal{A}$ might be too small and $\mathcal{P}$ is finite, so $\mathcal{A} \cap \mathcal{P}$ could be empty. Thus we analyze the misclassification rate under a relaxation of Assumption 2 (see Section 4).

## 3 Variational PU learning

### 3.1 Variational principle

In this section we establish a novel variational principle for PU learning without class prior estimation that will be used in rest of this paper. According to the Bayes rule, for a given parametric model $\Phi$ of the Bayesian classifier $\Phi^*$, the positive data distribution $f_P$ can be approximated by

$$
\begin{aligned}
f_P(x) &= \frac{\mathbb{P}(y = +1|x)\mathbb{P}(x)}{\int \mathbb{P}(y = +1|x)\mathbb{P}(x)\mathrm{d}x} \\
&\approx \frac{\Phi(x)f(x)}{\mathbb{E}_f[\Phi(x)]} \triangleq f_\Phi(x),
\end{aligned}
\tag{4}
$$

and we can further prove that $f_\Phi = f_P$ if and only if $\Phi = \Phi^*$ under Assumptions 1 and 2.[3] Then, the approximation quality of $\Phi$ can be evaluated by some divergence between $f_P$ and $f_\Phi$, e.g., the Kullback-Leibler (KL) divergence $\mathrm{KL}(f_P||f_\Phi)$. The above analysis leads to our main theorem:

**Theorem 3.** For all $\Phi : \mathbb{R}^d \mapsto [0, 1]$ with $\mathbb{E}_f[\Phi(x)] > 0$,

$$\mathrm{KL}(f_P||f_\Phi) = \mathcal{L}_{\mathrm{var}}(\Phi) - \mathcal{L}_{\mathrm{var}}(\Phi^*), \tag{5}$$

under Assumption 1, where

$$\mathcal{L}_{\mathrm{var}}(\Phi) \triangleq \log \mathbb{E}_f[\Phi(x)] - \mathbb{E}_{f_P}[\log \Phi(x)]. \tag{6}$$

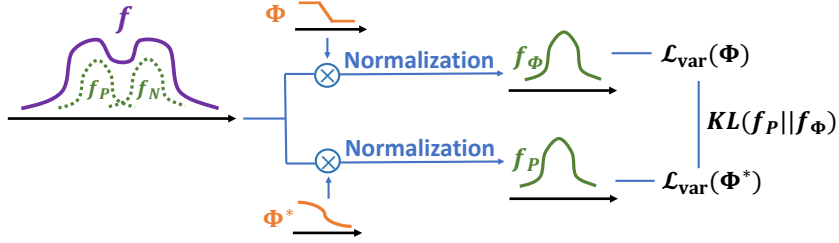

Figure 1: Graphical interpretation of the variational principle stated by Theorem 3, where $f_P, f_N, f$ denote distributions of positive, negative and unlabeled data. Each classifier model $\Phi$ induces an approximation $f_\Phi$ of $f_P$ as in (4), and $\mathrm{KL}(f_P||f_\Phi)$ equals to the difference between functionals $\mathcal{L}_{\mathrm{var}}(\Phi)$ and $\mathcal{L}_{\mathrm{var}}(\Phi^*)$.

Since the KL divergence is always nonnegative, $\mathcal{L}_{\mathrm{var}}(\Phi)$ provides a variational upper bound of $\mathcal{L}_{\mathrm{var}}(\Phi^*)$, which can be easily computed by empirical averages over sets $\mathcal{P}, \mathcal{U}$, and the KL divergence of $f_P$ from $f_\Phi$ can be minimized by equivalently minimizing $\mathcal{L}_{\mathrm{var}}(\Phi)$ (see Fig. 1 for illustration). As a result, by selecting a regularization functional $\mathcal{L}_{\mathrm{reg}}$ (see Section 3.2), parameters of $\Phi$ can be optimized by solving

$$\min_\Phi \mathcal{L}(\Phi) = \mathcal{L}_{\mathrm{var}}(\Phi) + \lambda \mathcal{L}_{\mathrm{reg}}(\Phi) \qquad (7)$$

subject to constraints $\Phi(x) \in [0, 1]$ and $\max_x \Phi(x) = 1$. In what follows, we refer to such a method variational PU (VPU) learning.

*Remark* 4. Theorem 3 can also be interpreted as a corollary to the Donsker-Varadhan representation theorem [23, 24] by utilizing the variational representation of $\mathrm{KL}(f_P||f)$. Based on the Donsker-Varadhan representation, objective functions similar to $\mathcal{L}_{var}$ have been proposed to tackle various problems, such as estimation of mutual information [24], density ratio estimation [25] and identification of information-leaking features [26].

*Remark* 5. Although $\mathcal{L}_{\mathrm{var}}$ is scalar invariant with $\mathcal{L}_{\mathrm{var}}(c \cdot \Phi) = \mathcal{L}_{\mathrm{var}}(\Phi)$ for $c > 0$ according to (6), $\Phi^*$ can be uniquely determined by the variational principle due to Assumption 2 (see Section A in Suppl. Material).

### 3.2 Regularized learning method

The variational principle provides an asymptotically correct way to model the classifier for PU learning in the limit of infinite sizes of $\mathcal{P}$ and $\mathcal{U}$. However, in many application scenarios, the size of labeled data is too small and the empirical distribution cannot represent $f_P$. Therefore, simply minimizing $\mathcal{L}_{\mathrm{var}}(\Phi)$ with a complex $\Phi$ may suffer from overfitting and yield underestimation of $\Phi^*(x)$ for positive but unlabeled data that are not close-neighbors of labeled data (see analysis in Section A of Suppl. Material).

To overcome the above-mentioned issue of non-robustness, we incorporate a *MixUp* [27] based consistency regularization term to the variational loss (7) as

$$\mathcal{L}_{\mathrm{reg}}(\Phi) = \mathbb{E}_{\tilde\Phi, \tilde x} \left[ \left( \log \tilde\Phi - \log \Phi(\tilde x) \right)^2 \right], \qquad (8)$$

with

$$\begin{aligned} \gamma &\overset{\mathrm{iid}}{\sim} \mathrm{Beta}(\alpha, \alpha), \\ \tilde x &= \gamma \cdot x' + (1 - \gamma) \cdot x'', \\ \tilde\Phi &= \gamma \cdot 1 + (1 - \gamma) \cdot \Phi(x''). \end{aligned} \qquad (9)$$

Here $\tilde x$ is a sample generated by mixing randomly selected $x' \in \mathcal{P}$ and $x'' \in \mathcal{U}$, and $\tilde\Phi$ represents the *guessed* probability $\mathbb{P}(y = +1|x = \tilde x)$ constructed by the linear interpolation of the true label and that predicted by $\Phi$. The consistency regularization is popular for semi-supervised learning methods [28, 29], and encourages smoothness of the model $\Phi$ especially in the area between labeled and unlabeled data in VPU. Unlike in [29], here we perform MixUp between labeled and unlabeled

---
**Algorithm 1** Stochastic gradient based VPU
---
1: **Input:** Positive and negative data sets $\mathcal{P}, \mathcal{U}$, a parametric model of $\Phi : \mathbb{R}^d \mapsto [0, 1]$, hyperparameters $\lambda$ and $\alpha$.
2: **repeat**
3:     Randomly sample mini-batches $\mathcal{B}^{\mathcal{P}}$ and $\mathcal{B}^{\mathcal{U}}$ from $\mathcal{P}$ and $\mathcal{U}$ with batch size $B$.
4:     Compute the variational loss by

$$\hat{\mathcal{L}}_{var} = \log \frac{\sum_{x \in \mathcal{B}^{\mathcal{U}}} \Phi(x)}{B} - \frac{\sum_{x \in \mathcal{B}^{\mathcal{P}}} \log \Phi(x)}{B}.$$

5:     Sample $\gamma \sim \text{Beta}(\alpha, \alpha)$, and perform MixUp between labeled and unlabeled data by for $i = 1, \cdots, B$

$$\tilde{x}_i = \gamma x_i^{\mathcal{P}} + (1 - \gamma) x_i^{\mathcal{U}}$$
$$\tilde{\Phi}_i = \gamma + (1 - \gamma) \Phi(x_i^{\mathcal{U}}),$$

    where $x_i^{\mathcal{P}} \in \mathcal{B}^{\mathcal{P}}, x_i^{\mathcal{U}} \in \mathcal{B}^{\mathcal{U}}$.
6:     Compute the regularization term and the total loss by

$$\hat{\mathcal{L}}_{\text{reg}} = \frac{1}{B} \sum_{i=1}^{B} \left[ \log \Phi(\tilde{x}_i) - \log \tilde{\Phi}_i \right]^2,$$
$$\hat{\mathcal{L}} = \hat{\mathcal{L}} + \lambda \hat{\ell}_{\text{reg}}.$$

7:     Update parameters $W$ of $\Phi$ with step-sizes $\eta$ as

$$W \leftarrow W - \eta \frac{\partial \hat{\mathcal{L}}}{\partial W}.$$

8: **until** The terminal condition is satisfied.
9: Perform the normalization

$$\Phi(x) \leftarrow \min \left\{ \frac{\Phi(x)}{\max_{x \in \mathcal{P} \cup \mathcal{U}} \Phi(x)}, 1 \right\}$$

    according to Remark 5.
---

samples, and quantify the consistency between the predicted and interpolated $\Phi(\tilde{x})$ by the mean squared logarithmic error rather than the mean squared error used in [29], because this scheme penalizes more heavily the underestimation of $\Phi(\tilde{x})$ (see Section A in Suppl. Material for detailed analysis). The effectiveness of the proposed consistency regularization is validated by our ablation study in Section 5.4. Finally, it is noteworthy that some other regularization schemes without data augmentation can also work well in the VPU framework (see, e.g., Section B.7).

A stochastic gradient based implementation of VPU with loss function defined by (7) and (8) is given in Algorithm 1, where regularization parameters $\lambda$ and $\alpha$ can be tuned by comparing the variational loss $\mathcal{L}_{\text{var}}(\Phi)$ on the validation set.

## 3.3 Comparison with related work

From an algorithmic perspective, VPU is similar to the risk estimator based PU learning methods, uPU and nnPU [10, 12]. All the three methods optimize parameters of the classifier with respect to some empirical estimates of a loss under the SCAR assumption, and the major difference comes from the fact that by introducing Assumption 2, VPU can be implemented without the class prior $\pi_P$. As analyzed in Section 2 and Section A in Suppl. Material, Assumption 2 comprises most practical cases where some instances are positive with probability one, and most class prior estimation methods require similar irreducibility assumptions for the identifiability of $\pi_P$. It can also be proved that a slight relaxation of this assumption will not significantly affect the asymptotic correctness of VPU (see Theorem 8).

Furthermore, all hyperparameters in VPU, including regularization parameters, model class and iteration number, can be determined by $\mathcal{L}_{\text{var}}$ based cross validation. But for uPU and nnPU, because the estimated risks heavily rely on the class prior $\pi_P$, choosing $\pi_P$ by the direct estimated risk based cross validation will yield an uninformative result with $\pi_P = 1$. (See analysis in Section A of Suppl. Material.)

## 4 Theoretical analysis

The asymptotic correctness of VPU is a direct consequence of the variational principle introduced in Section 3.1 as shown in the following theorem.

**Theorem 6.** *Provided that the following conditions hold: (i) Assumptions 1 and 2 hold, (ii) and the classifier is modeled as $\Phi(x) = \Phi(x, \theta)$ with parameters $\theta$ and there exists $\theta^*$ so that $\Phi^*(x) = \Phi(x, \theta^*)$. Then the optimal $\Phi$ obtained by VPU satisfies $\Phi \xrightarrow{p} \Phi^*$ as $M, N \to \infty$ and $\lambda \to 0$.*

We now analyze the effects of relaxation of Assumptions 1 and 2 on VPU.

**Assumption 7.** $\mathcal{P} \overset{\text{iid}}{\sim} f'_P, \mathcal{U} \overset{\text{iid}}{\sim} f$, where $f'_P$ differs from the positive data distribution $f_P$ in $\mathcal{U}$, and $f'_P, f_P$ satisfy (i) there are positive constants $c_1, c_2$ close to 1 so that $c_1 f_P(x) \leq f'_P \leq c_2 f_P(x)$ and (ii) there is a set $\mathcal{A} \subset \mathbb{R}^d$ with $\int_{\mathcal{A}} f_P(x)\mathrm{d}x > 0$ so that $\min_{x \in \mathcal{A}} \Phi^*(x) \geq 1 - \epsilon$ with $\epsilon \in [0, 1)$ being a small number.

**Theorem 8.** *If data distributions satisfy Assumption 7, the optimal $\Phi$ obtained by VPU with $\lambda = 0$ and $M, N \to \infty$ satisfies*

$$|\mathcal{R}(\Phi) - \mathcal{R}(\Phi^*)| \leq \max\left\{\frac{c_2}{c_1} - 1, 1 - \frac{c_1(1-\epsilon)}{c_2}\right\},$$

*where $\mathcal{R}(\Phi)$ denotes the misclassification rate of the predicted label $y = \mathrm{sign}(\Phi(x) - 0.5)$.*

Selection bias is a practically important but theoretically challenging classification problem for VPU, which implies that the labeled data distribution $f'_P$ may differ from the positive data distribution $f_P$ [30, 31]. Although the variational principle in this case requires further investigations, Theorem 8 ensures that the VPU learning can still obtain a classification accuracy comparable to the ideal $\Phi^*$, i.e., $\mathcal{R}(\Phi) \approx \mathcal{R}(\Phi^*)$, if the selection bias is limited with $c_1, c_2$ close to 1 and Assumption 7 is only slightly violated with $\epsilon \ll 1$. Our numerical experiments also indicate that the proposed VPU is quite robust to the bias of labeled data (see Section 5.5).

## 5 Experiments

In this section, we test the effectiveness of VPU on both synthetic and real-world datasets. We provide an extensive ablation study to analyze the regularization defined by (8). Considering selection bias is common in practice, we test the effectiveness of VPU and existing methods in this scenario. At last, we further demonstrate the robustness of VPU by experiments with different size of the labeled set.

### 5.1 Implementation details

The class label is predicted as $y = \mathrm{sign}(\Phi(x) - 0.5)$ in VPU when calculating classification accuracies. In all experiments, $\alpha$ is chosen as $0.3$ and $\lambda \in \{1e-4, 3e-4, 1e-3, \cdots, 1, 3\}$ is determined by holdout validation unless otherwise specified. We use Adam as the optimizer for VPU with hyperparameters $(\beta_1, \beta_2) = (0.5, 0.99)$.

The performance of VPU is compared to that of some recently developed PU learning methods, including the unbiased risk estimator based uPU and nnPU [10, 12], the generative model based GenPU [14], and the rank pruning (RP) proposed in [9].[4] Notice that uPU and nnPU require the prior knowledge of the class proportion. Thus, for fair comparison, $\pi_P$ is estimated by the KM2 method proposed in [32] when implementing uPU and nnPU, where KM2 is one of the state-of-the-art class

prior estimation algorithms. For GenPU, the hyperparameters of the algorithm are determined by greedy grid search as described in Section B in Suppl. Material.

In all the methods, the classifiers (including discriminators of GenPU) are modeled by 7-layer MLP for UCI datasets, LeNet-5 [33] for FashionMNIST and 7-layer CNN for CIFAR-10 and STL-10. By default, the accuracies are evaluated on test sets and the mean and standard deviation values are computed from 10 independent runs. All the other detailed settings of datasets and algorithms are provided in Section B of Suppl. Material, and the software code for VPU is also available[5].

## 5.2  Benchmark data

We conduct experiments on three benchmark datasets taken from the UCI Machine Learning Repository [34, 35], and the classification results are reported in Table 2. It can be seen that VPU outperforms the other methods with high accuracies and low variances on almost all the datasets. nnPU and uPU suffer from the estimation error of $\pi_P$. In fact, if $\pi_P$ is exactly given, nnPU can achieve better performance, though still a little worse than VPU. (See Section B in Suppl. Material.) In addition, RP interprets unlabeled data as noisy negative data and can get an accurate classifier when the proportion of positive data is small in unlabeled data. But in the opposite case where the proportion is too large, RP performs even worse than random guess. ($\pi_P = 0.896$ and $0.635$ in Page Blocks with 'text' vs 'horizontal line, vertical line, picture, graphic and Grid Stability with 'unstable' vs 'stable'.)

Table 2: Classification accuracies (%) of compared methods on UCI datasets. Definitions of labels ('Positive' vs 'Negative') are as follows: Page Blocks[1]: 'horizontal line , vertical line, picture, graphic' vs 'text'. Page Blocks[2]: 'text' vs 'horizontal line , vertical line, picture, graphic'. Grid Stability[1]: 'stable' vs 'unstable'. Grid Stability[2]: 'unstable' vs 'stable'. Avila[1]: 'A' vs the rest. Avila[2]: 'A, F' vs the rest. Labeled positive data are randomly selected from the training data with $M = 100, 1000, 2000$ and $N = 3284, 6000, 10430$.

| Dataset | Page Blocks[1] | Page Blocks[2] | Grid Stability[1] | Grid Stability[2] | Avila[1] | Avila[2] |
|---|---|---|---|---|---|---|
| VPU | $\mathbf{93.6 \pm 0.4}$ | $\mathbf{93.5 \pm 0.7}$ | $\mathbf{92.6 \pm 0.3}$ | $\mathbf{89.5 \pm 0.5}$ | $\mathbf{82.0 \pm 0.9}$ | $\mathbf{87.2 \pm 0.5}$ |
| nnPU | $93.4 \pm 1.1$ | $90.2 \pm 2.6$ | $80.8 \pm 2.5$ | $84.1 \pm 1.8$ | $73.3 \pm 2.0$ | $83.1 \pm 2.1$ |
| uPU | $92.8 \pm 1.3$ | $86.8 \pm 4.7$ | $\mathbf{92.6 \pm 0.7}$ | $86.8 \pm 0.5$ | $75.0 \pm 0.4$ | $82.7 \pm 1.7$ |
| GenPU | $93.2 \pm 0.3$ | $90.2 \pm 0.1$ | $69.3 \pm 0.6$ | $75.6 \pm 1.8$ | $63.4 \pm 1.1$ | $67.1 \pm 0.8$ |
| RP | $91.2 \pm 1.4$ | $9.96 \pm 0.7$ | $84.7 \pm 1.3$ | $36.7 \pm 0.6$ | $75.8 \pm 0.4$ | $77.2 \pm 0.2$ |

## 5.3  Image datasets

Here we compare all the methods on three image datasets: FashionMNIST, CIFAR-10, and STL-10. Notice that in the rest of the paper, we denote the 10 classes of each image datasets with integers ranging from 0 to 9, following the default settings in torchvision 0.5.0 (see Section B in Suppl. Material).[6] The classification accuracies are collected in Table 3, in which the superiority of VPU is also marked (see Section B.8 for other comparison metric). Here uPU performs much worse than nnPU due to the overfitting problem [12]. Moreover, the performance of GenPU is also not satisfying because of the mode collapse of generators (see Section B in Suppl. Material).

## 5.4  Ablation study

To justify our choice for the regularization term (8), we conduct an ablation study on FashionMNIST with '1, 4, 7' as positive labels and 1000 labeled samples. We compare (a) consistency regularization (8) adopted in this paper with $x' \in \mathcal{P}$ and $x'' \in \mathcal{U}$ as in (9), (b) $\mathcal{L}_{\mathrm{reg}}(\Phi) \equiv 0$, (c) regularization with MixUp on $\mathcal{P}$ data only, (d) regularization with MixUp on $\mathcal{P} \cup \mathcal{U}$, where $x', x''$ are both randomly

Table 3: Classification accuracies (%) of compared methods on FashionMNIST (abbreviated as "F-MNIST"), CIFAR-10 and STL-10 datasets. Definitions of labels ('Positive' vs 'Negative') are as follows: FashionMNIST[1]: '1,4,7' vs '0,2,3,5,6,8,9'. FashionMNIST[2]: '0,2,3,5,6,8,9' vs '1,4,7'. CIFAR-10[1]: '0,1,8,9' vs '2,3,4,5,6,7'. CIFAR-10[2]: '2,3,4,5,6,7' vs '0,1,8,9'. STL-10[1]: '0,2,3,8,9' vs '1,4,5,6,7'. STL-10[2]: '1,4,5,6,7' vs '0,2,3,8,9'. For FashionMNIST and CIFAR-10, labeled positive data are randomly selected from the training data with $M = 3000$. For STL-10, $\mathcal{P}$ are defined as all positive labeled data in the training set with $M = 2500$.

| Dataset | F-MNIST[1] | F-MNIST[2] | CIFAR-10[1] | CIFAR-10[2] | STL-10[1] | STL-10[2] |
|---------|------------|------------|-------------|-------------|-----------|-----------|
| VPU | $\mathbf{92.7 \pm 0.3}$ | $\mathbf{90.8 \pm 0.6}$ | $\mathbf{89.5 \pm 0.1}$ | $\mathbf{88.8 \pm 0.8}$ | $\mathbf{79.7 \pm 1.5}$ | $\mathbf{83.7 \pm 0.1}$ |
| nnPU | $90.8 \pm 0.6$ | $90.5 \pm 0.4$ | $85.6 \pm 2.3$ | $85.5 \pm 2.0$ | $78.3 \pm 1.2$ | $82.2 \pm 0.5$ |
| uPU | $89.9 \pm 1.0$ | $78.6 \pm 1.3$ | $80.6 \pm 2.1$ | $72.9 \pm 3.2$ | $70.3 \pm 2.0$ | $74.0 \pm 3.0$ |
| Genpu | $47.8 \pm 1.0$ | $78.8 \pm 0.3$ | $67.6 \pm 0.9$ | $72.1 \pm 1.1$ | $65.1 \pm 1.0$ | $68.1 \pm 1.3$ |
| RP | $92.2 \pm 0.4$ | $75.9 \pm 0.6$ | $86.7 \pm 2.9$ | $77.8 \pm 2.5$ | $67.8 \pm 4.6$ | $68.5 \pm 5.7$ |

Table 4: Ablation study results on FashionMNIIST with '1, 4, 7' as positive labels and 1000 labeled samples. (8) is the regularization term we adopt, i.e., mean squared logarithmic error with MixUp between $\mathcal{P}$ and $\mathcal{U}$.

| Ablation on regularization | (8) | no regularization | (8) with MixUp on $\mathcal{P}$ only | (8) with MixUp on $\mathcal{P} \cup \mathcal{U}$ | (8) with MSE |
|---|---|---|---|---|---|
| Test accuracies | $91.3 \pm 0.4$ | $87.2 \pm 2.9$ | $90.3 \pm 0.3$ | $90.1 \pm 0.9$ | $90.0 \pm 0.2$ |

selected from $\mathcal{P} \cup \mathcal{U}$, (e) consistency loss defined by the mean squared error $\mathbb{E}_{\tilde{\Phi}, \tilde{x}}\left[\left(\tilde{\Phi} - \Phi(\tilde{x})\right)^2\right]$. Results in Table 4 show the superiority of (8).

## 5.5 Selection bias

In many practical situations, the assumption that the empirical distribution of $f_P$ is consistent with the ground truth may not be satisfied. Hence, in this section we compare the PU methods in Section 5.2 on FashionMNIST with '1, 4, 7' as positive labels under selection bias of $\mathcal{P}$. In this experiment, the total number of labeled data is fixed to 3000, but selection among different positive labels is biased. For positive labels '1, 4, 7', we denote corresponding numbers of labeled data as $n_1, n_4, n_7$, which satisfy $n_1 + n_4 + n_7 = 3000$ and $n_4 = n_7 \leq n_1$. (Note the three classes have the same size in the whole data set.) Performance of the methods is compared in Fig. 2, which shows that VPU has a superior robustness to sample selection bias of $\mathcal{P}$ over other methods. Poor performance of nnPU is, to a large extent, attributed to the difficulties of class prior estimation under selection bias, and nnPU is robust if the accurate class prior is known (See Section B in Suppl. Material).

## 5.6 Different size of the labeled set

Considering that a big labeled positive set is usually inaccessible in applications, we investigate performance of the PU methods with small labeled set on FashionMNIST with '1, 4, 7' as positive labels. The labeled set size ranges from 500 to 3000, and Figure 3 shows the robustness of VPU.

## 6 Conclusion

In this work, we proposed a novel variational principle for PU learning, and developed an efficient learning method called variational PU (VPU). In addition, a MixUp based regularization was utilized to improve the stability of the method. We also showed that the method can consistently estimate the optimal Bayesian classifier under a general condition without any assumption on class prior or data separability. The superior performance and robustness of VPU was confirmed in the experiments.

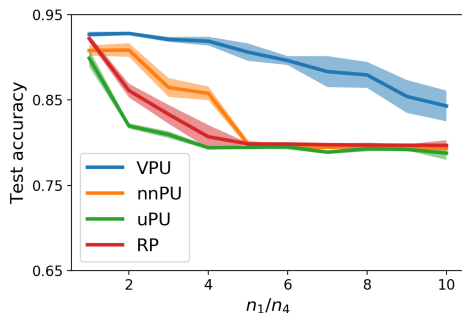
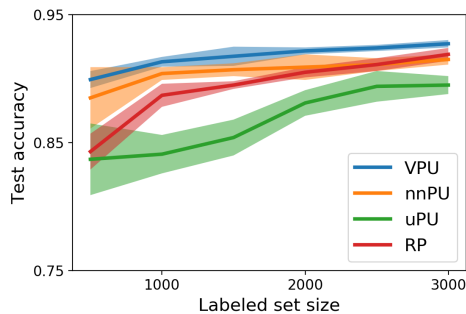

Figure 2: Test accuracies of PU methods on FashionMNIST with '1, 4, 7' as positive labels. $n_1, n_4, n_7$ denote corresponding numbers of labeled samples for each label, with $n_1+n_4+n_7 = 3000$ and $n_4 = n_7$.

Figure 3: Test accuracies of PU methods on FashionMNIST with '1, 4, 7' as positive labels, with different size of the labeled set.

It is worthy to note that variational principle could be extended to a more general framework by using different statistical distances, and some other possible variational principles are discussed in Section C of Suppl. Material. Many advanced techniques developed for measuring difference between distributions for GAN can be expected to improve the performance of VPU.

## Broader Impact

VPU is a general framework for PU learning, and it overcomes some limitations of previous methods, including requirement of class prior known beforehand and data separability, so is more applicable to real-world applications. Thus discussion of the potential impacts of VPU actually leads to the discussion of potential impacts of applications of PU learning itself. With VPU, less labels are needed, which saves cost and improves efficiency. Moreover, VPU is able to mine the negative pattern that is missing in the PU datasets. This will be helpful if finding out the negative pattern is beneficial, such as discovering drugs for diseases and identifying deceptive reviews for recommendation systems. However, malicious tasks can also be conducted with VPU, such as discovery of harmful chemical substance. Another unethical scenario is that sometimes the negative pattern could be hidden on purpose for the sake of privacy or other ethical considerations, but with VPU, people might be able to find out about the hidden information.

## Acknowledgments and Disclosure of Funding

The authors thank the anonymous NeurIPS reviewers for their valuable feedback. Hao Wu is supported by the Fundamental Research Funds for the Central Universities, China (No. 22120200276). Yin Wang is supported by National Natural Science Foundation of China (No. 61950410614) and Cross-disciplinary Program for the Central Universities, China (No. 08002150042).

## Footnotes

[3]All proofs can be found in Section A in Suppl. Material.

[4]The software codes are downloaded from `https://github.com/kiryor/nnPUlearning`, `https://qibinzhao.github.io/index.html` and `https://github.com/cgnorthcutt/rankpruning`.

[5]https://github.com/HC-Feynman/vpu

[6]Datasets are downloaded from https://github.com/zalandoresearch/fashion-mnist, https://www.cs.toronto.edu/~kriz/cifar.html and http://cs.stanford.edu/~acoates/stl10.

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
