[Supplementary Material · Supplementary Material.pdf]

# Supplementary Material

## A Analysis of VPU

### A.1 Class prior estimation in uPU and nnPU

The risk estimator in uPU [10, 11] is defined by (1), and nnPU [12] provides a nonnegative estimator

$$\mathrm{risk}(\Phi) = \pi_P \mathbb{E}_{\text{labeled data}} \left[\ell_+ \left(\Phi(x)\right)\right] + \max\left\{0, \mathbb{E}_{\text{unlabeled data}} \left[\ell_- \left(\Phi(x)\right)\right] - \pi_P \mathbb{E}_{\text{labeled data}} \left[\ell_- \left(\Phi(x)\right)\right]\right\}$$
(10)

in order to avoid overfitting, where the classifier $\Phi$ is not necessarily an approximate Bayesian classifier and its range can be $\mathbb{R}$. Both (1) and (10) consistently estimate the misclassification risk

$$\pi_P \mathbb{E}_{f_P} \left[\ell_+ \left(\Phi(x)\right)\right] + (1 - \pi_P)\mathbb{E}_{f_N} \left[\ell_- \left(\Phi(x)\right)\right]$$

under Assumption 1, where $f_N$ denotes negative distribution $\mathbb{P}(x|y = -1)$. In usual cases, loss functions $\ell_+$ and $\ell_-$ satisfy [12]

1. $\ell_+ \left(\Phi(x)\right) \geq 0$ and $\ell_- \left(\Phi(x)\right) \geq 0$ for all $x$.

2. $\ell_+ \left(\Phi(x)\right) \to 0$ as $\Phi(x) \to C$ for some constant $C$, where $C$ can be $\infty$. This implies the loss is zero if $\Phi$ classify a positive sample $x$ as positive with a high confidence.

If we minimize the estimated risk by regarding $\pi_P$ as a variable, a trivial minimum of $0$ can be achieved with $\pi_P = 1$ and $\Phi(x) \equiv C$ in the limit of infinite data size, i.e., all data are predicted as positive. achieves a trivial minimum of 0 with $\pi_P = 1$ and $\Phi(x) \equiv C$, i.e., unlabeled data are predicted as positive. This result is obviously uninformative. Moreover, it is also infeasible to select $\pi_P$ as a hyperparameter by the estimated risk based cross validation, since the minimal estimated risk on validation set can also be obtained with $\pi_P = 1$ and $\Phi(x) \equiv C$. Therefore, unless some negative samples are available as validation data, the class prior estimation is an unavoidable intermediate step when performing uPU or nnPU.

### A.2 Assumption 2 and irreducibility assumption

According to Assumption 1, the unlabeled data distribution can be decomposed as

$$f = \pi_P \cdot f_P + (1 - \pi_P) \cdot f_N,$$

where $f_N = \mathbb{P}(x|y = -1)$, and it can be rewritten as

$$f = \pi_P' \cdot f_P + (1 - \pi_P')f_N'$$

with

$$\pi_P' = c\pi_P,$$
$$f_N' = \frac{(1 - c)\pi_P \cdot f_P + (1 - \pi_P) \cdot f_N}{1 - c\pi_P},$$

for all $c \in (0, 1)$. This implies that $f_N$ and $\pi_P$ cannot be uniquely determined from $f, f_P$ if $f_N$ is a mixture distribution which contains $f_P$. In order to deal with this problem, most class prior estimation methods [16–19] assume that $f_N$ is irreducible with respect to $f_P$, i.e., if $f_N$ is not a mixture containing $f_P$ [33]. One stronger variant of the irreducibility assumption is [34, 35]

$$\min_{\mathcal{A} \subset \mathbb{R}^d, \int_{\mathcal{A}} f_P(x)\mathrm{d}x > 0} \frac{f_N(x)}{f_P(x)} = 0.$$
(11)

We now show that Assumption 2 is equivalent to (11).

**Proposition 9.** *Assumption 2 is satisfied if and only if (11) holds.*

*Proof.* If (11) holds and $\mathcal{A}$ is an optimal solution,

$$
\begin{aligned}
\Phi^*(x) &= \frac{\pi_P f_P(x)}{f(x)} \\
&= \frac{\pi_P f_P(x)}{\pi_P f_P(x) + (1 - \pi_P) f_N(x)} \\
&= \frac{1}{1 + \frac{1 - \pi_P}{\pi_P} \frac{f_N(x)}{f_P(x)}} \\
&= 1
\end{aligned}
$$

for all $x \in \mathcal{A}$, and therefore Assumption 2 is satisfied by $\mathcal{A}$. If Assumption 2 holds with set $\mathcal{A}$,

$$
\begin{aligned}
\frac{f_N(x)}{f_P(x)} &= \frac{f(x) - \pi_P f_P(x)}{(1 - \pi_P) f_P(x)} \\
&= \frac{f(x) - f(x)\Phi^*(x)}{(1 - \pi_P) f_P(x)} \\
&= 0,
\end{aligned}
$$

for all $x \in \mathcal{A}$, which implies that (11) also holds. Proof of Theorem 3 □

According to (4) and the definition of KL divergence,

$$
\begin{aligned}
\mathrm{KL}(f_P \| f_\Phi) &= \mathbb{E}_{f_P} \left[ \log \frac{f_P(x)}{f_\Phi(x)} \right] \\
&= \mathbb{E}_{f_P} \left[ \log \Phi^*(x) \right] + \mathbb{E}_{f_P} \left[ \log f(x) \right] - \log \mathbb{E}_f \left[ \Phi^*(x) \right] \\
&\quad - \left( \mathbb{E}_{f_P} \left[ \log \Phi(x) \right] + \mathbb{E}_{f_P} \left[ \log f(x) \right] - \log \mathbb{E}_f \left[ \Phi(x) \right] \right) \\
&= -\mathcal{L}_{\mathrm{var}} \left( \Phi^* \right) + \mathcal{L}_{\mathrm{var}} \left( \Phi \right).
\end{aligned}
$$

### A.3 Analysis of minimum points of $\mathcal{L}_{\mathrm{var}}$

**Proposition 10.** *For any constant $c > 0$, $\mathcal{L}_{\mathrm{var}} (c \cdot \Phi) = \mathcal{L}_{\mathrm{var}} (\Phi)$.*

*Proof.* From the definition of $\mathcal{L}_{\mathrm{var}}$, we get

$$
\begin{aligned}
\mathcal{L}_{\mathrm{var}} (c \cdot \Phi) &= \log \mathbb{E}_f \left[ c \cdot \Phi(x) \right] - \mathbb{E}_{f_P} \left[ \log (c \cdot \Phi(x)) \right] \\
&= \log \int c f(x) \Phi(x) \mathrm{d}x - \int f_P(x) \log (c \cdot \Phi(x)) \, \mathrm{d}x \\
&= \log c + \log \int f(x) \Phi(x) \mathrm{d}x \\
&\quad - (\log c) \cdot \int f_P(x) \mathrm{d}x - \int f_P(x) \log \Phi(x) \mathrm{d}x \\
&= \log \int f(x) \Phi(x) \mathrm{d}x - \int f_P(x) \log \Phi(x) \mathrm{d}x \\
&= \mathcal{L}_{\mathrm{var}} (\Phi)
\end{aligned}
$$

for any $c > 0$. □

It can be seen from the above proposition that $\Phi^*$ is not the unique minimum point of $\mathcal{L}_{\mathrm{var}}$. For all $c \in (0, \frac{1}{\sup_x \Phi^*(x)}]$, $\Phi = c \cdot \Phi^*$ satisfies

$$
0 < \Phi(x) \le \frac{\Phi^*(x)}{\sup_x \Phi^*(x)} \le 1
$$

and is also a minimum point. In fact, we can show that all minimum points of $\mathcal{L}_{\mathrm{var}}$ are in the form of $\Phi = c \cdot \Phi^*$.

**Proposition 11.** *A function $\Phi : \mathbb{R}^d \mapsto [0, 1]$ satisfies $\mathcal{L}_{\mathrm{var}}(\Phi) = \mathcal{L}_{\mathrm{var}}(\Phi^*)$ iff $\Phi = c \cdot \Phi^*$ and $c \in (0, \frac{1}{\sup_x \Phi^*(x)}]$.*

*Proof.* The sufficiency is trivial, and so we only give the proof of the necessity. Suppose that $\Phi$ is a minimum point of $\mathcal{L}_{\text{var}}$, then

$$
\begin{aligned}
\frac{\partial}{\partial \epsilon} \mathcal{L}_{\text{var}}(\Phi + \epsilon h)\bigg|_{\epsilon=0} &= \frac{\partial}{\partial \epsilon} \log \int f(x)\,(\Phi(x) + \epsilon h(x))\,\mathrm{d}x \\
&\quad - \frac{\partial}{\partial \epsilon} \int f_P(x) \log\,(\Phi(x) + \epsilon h(x))\,\mathrm{d}x \\
&= \int \frac{f(x)h(x)}{\mathbb{E}_f[\Phi]} - \frac{f_P(x)h(x)}{\Phi(x)}\,\mathrm{d}x
\end{aligned}
$$

must be zero for an arbitrary function $h(x)$. Hence,

$$
\frac{f(x)}{\mathbb{E}_f[\Phi]} - \frac{f_P(x)}{\Phi(x)} \equiv 0
$$

$$
\Rightarrow \Phi(x) \equiv \mathbb{E}_f[\Phi]\frac{f_P(x)}{f(x)}
$$

By combining the above equation and the Bayes rule, we have

$$
\begin{aligned}
\Phi(x) &= \mathbb{E}_f[\Phi]\frac{f_P(x)}{f(x)} \\
&= \frac{\mathbb{E}_f[\Phi]}{\pi_P} \cdot \Phi^*(x).
\end{aligned}
$$

It is obvious that $c = \frac{\mathbb{E}_f[\Phi]}{\pi_P} > 0$. In addition, we can conclude from $\sup_x \Phi(x) \leq 1$ that

$$
\sup_x c \cdot \Phi^*(x) \leq 1 \Rightarrow c \leq \frac{1}{\sup_x \Phi^*(x)}.
$$

$\square$

Based on the above analysis, we can conclude that $\Phi^*$ can be uniquely determined for given $f$ and $f_P$ under Assumption 2.

**Proposition 12.** *If Assumption 2 is satisfied and $\sup_x \Phi(x) = 1$ for a function $\Phi : \mathbb{R}^d \mapsto [0, 1]$, $\mathcal{L}_{\text{var}}(\Phi) = \mathcal{L}_{\text{var}}(\Phi^*)$ iff $\Phi = \Phi^*$.*

*Proof.* This is a trivial corollary of Proposition 11. $\square$

Furthermore, the following proposition provides the optimal solutions in the case where only estimated $f, f_P$ are available.

**Proposition 13.** *All solutions to $\min_\Phi \hat{\mathcal{L}}_{\text{var}}(\Phi)$ with $\hat{\mathcal{L}}_{\text{var}}(\Phi) = \log \mathbb{E}_{\hat{f}}[\Phi(x)] - \mathbb{E}_{\hat{f}_P}[\log(\Phi(x))]$ satisfy*

$$
\Phi(x) \propto \hat{f}_P(x)/\hat{f}(x).
$$

*Proof.* Omitted as it is similar to that of Proposition 11. $\square$

Analysis of Regularization

For given $\mathcal{P}$ and $\mathcal{U}$, the empirical estimate of $\mathcal{L}_{\text{var}}(\Phi)$ is

$$
\hat{\mathcal{L}}_{\text{var}}(\Phi) = \log \frac{1}{N} \sum_{x \in \mathcal{U}} \Phi(x) - \frac{1}{M} \sum_{x \in \mathcal{P}} \log \Phi(x).
$$

Therefore, if the capacity of the model $\Phi$ is extremely high, simply minimizing $\hat{\mathcal{L}}_{\text{var}}(\Phi)$ yields

$$
\Phi(x) = \begin{cases} 1, & x \in \mathcal{P}, \\ 0, & \text{otherwise.} \end{cases} \tag{12}
$$

This overfitting issue can be partly alleviated by early stopping, i.e., stopping the training when $\mathcal{L}_{\mathrm{var}}(\Phi)$ estimated on the validation set starts to increase. But according to our numerical experience, it can be more effectively overcome by the *MixUp* based regularization described in Section 3.2.

For two randomly selected $x' \in \mathcal{P}$ and $x'' \in \mathcal{U}$, if $\Phi^*(\tilde{x})$ is extremely underestimated with $\Phi(\tilde{x}) \to 0$ for the virtual sample $\tilde{x}$ (see (9)), we can conclude that the regularization w.r.t. $\tilde{x}$

$$
\begin{aligned}
\left( \log \tilde{\Phi} - \log \Phi(\tilde{x}) \right)^2 &\geq (\log \gamma - \log \Phi(\tilde{x}))^2 \\
&= \mathcal{O}\left( (\log \Phi(\tilde{x}))^2 \right) \to \infty
\end{aligned}
$$

as $\Phi(\tilde{x}) \to 0$. Thus, with the regularization (8), the resulting $\Phi(x)$ decay smoothly outside of $\mathcal{P}$ and the trivial solution (12) is excluded.

Another possible choice is the mean square error based regularization $\mathbb{E}_{\tilde{\Phi}, \tilde{x}}\left[ \left( \tilde{\Phi} - \Phi(\tilde{x}) \right)^2 \right]$, but this regularization term is bounded and penalizes less for overfitting.

We can also define the regularization by using the standard cross-entropy loss, which yields the regularization loss

$$
-\tilde{\Phi} \log \Phi(\tilde{x}) - \left( 1 - \tilde{\Phi} \right) \log \left( 1 - \Phi(\tilde{x}) \right) = \mathcal{O}\left( -\log \Phi(\tilde{x}) \right)
$$

for each $\tilde{x}$. It can be seen that the proposed mean squared logarithmic error based regularization penalizes more heavily the underestimation of $\Phi(\tilde{x})$.

Another possible choice is the mean square error based regularization $\mathbb{E}_{\tilde{\Phi}, \tilde{x}}\left[ \left( \tilde{\Phi} - \Phi(\tilde{x}) \right)^2 \right]$, but this regularization term is bounded and penalizes less for overfitting has less penalization.

As for the MixUp strategy, the MixUp between $\mathcal{P}$ and $\mathcal{U}$ ensures that $\tilde{\Phi} \approx 1$ for $\gamma \approx 1$ and $\tilde{\Phi} > \Phi(x'')$ (see (9)), so it can solve the overfitting problem by penalizing the underestimation of $\Phi(x)$ heavily for unlabeled data. As a comparison, MixUp inside $\mathcal{P}$ or $\mathcal{U}$ cannot effectively penalize the underestimation of $\Phi(x)$ outside of $\mathcal{P}$. So we implement MixUp between $\mathcal{P}$ and $\mathcal{U}$ as in (9), and can lead to more accurate and robust classifier according to our numerical experience than MixUp on $\mathcal{P} \cup \mathcal{U}$ (i.e., $x'$ and $x''$ are both randomly drawn from $\mathcal{P} \cup \mathcal{U}$) according to our numerical experience.

The advantage of (8) is demonstrated in Section 5.4.

### A.4 Proof of Theorem 6

Notice that the variational loss estimated from data

$$
\begin{aligned}
\hat{\mathcal{L}}_{\mathrm{var}}(\Phi(\cdot, \theta)) &= \log \frac{\sum_{x \in \mathcal{U}} \Phi(x, \theta)}{N} - \frac{\sum_{x \in \mathcal{P}} \log \Phi(x, \theta)}{M} \\
&\xrightarrow{p} \mathcal{L}_{\mathrm{var}}(\Phi(\cdot, \theta))
\end{aligned}
$$

for a given $\theta$ as $M, N \to \infty$. According to Theorem 2.1 in [36] and Proposition 12, we can conclude that the optimal solution $\Phi(x, \theta)$ to (7) converges to $\Phi(x, \theta^*)$ when $M, N \to \infty$ and $\lambda \to 0$.

### A.5 Proof of Theorem 8

By considering condition (ii) in Assumption 7 and the fact that $\Phi^*(x)$ can be written as

$$
\Phi^*(x) = Z^{-1} f_P(x) / f(x),
$$

we have

$$
\begin{aligned}
\max_x \Phi^*(x) &= Z^{-1} \max_x f_P(x) / f(x) \in [1 - \epsilon, 1] \\
\Rightarrow Z &\in \left[ \max_x f_P(x) / f(x), \frac{\max_x f_P(x) / f(x)}{1 - \epsilon} \right].
\end{aligned}
$$

It can then be known from Proposition 13 that the optimal solution $\Phi$ to

$$\min \mathcal{L}'_{\text{var}}(\Phi) = \log \mathbb{E}_f[\Phi(x)] - \mathbb{E}_{f'_P}[\log \Phi(x)]$$

under constraint $\max_x \Phi(x) = 1$ is given by

$$\Phi(x) = \frac{f'_P(x)/f(x)}{\max_x f'_P(x)/f(x)}.$$

We can obtain from condition (i) in Assumption 7 that

$$
\begin{aligned}
\Phi(x) & \geq \frac{c_1 f_P(x)/f(x)}{c_2 \max f_P(x)/f(x)} \\
& \geq \frac{c_1 f_P(x)/f(x)}{c_2 Z} \\
& = \frac{c_1}{c_2} \Phi^*(x)
\end{aligned}
$$

and

$$
\begin{aligned}
\Phi(x) & \leq \frac{c_2 f_P(x)/f(x)}{c_1 \max f_P(x)/f(x)} \\
& \leq \frac{c_2 f_P(x)/f(x)}{c_1(1-\epsilon)Z} \\
& = \frac{c_2}{c_1(1-\epsilon)} \Phi^*(x)
\end{aligned}
$$

For convenience of analysis, we denote the misclassification probability of $\Phi$ for a given sample $x$ by

$$
\mathcal{R}_x(\Phi) = \begin{cases} \mathbb{P}\left(y = +1 | x\right), & \text{if } \Phi(x) < 0.5 \\ \mathbb{P}\left(y = -1 | x\right), & \text{if } \Phi(x) \geq 0.5 \end{cases}.
$$

Thus,

$$
\begin{aligned}
\mathcal{R}_x(\Phi) - \mathcal{R}_x(\Phi^*) & = \Phi^*(x) \cdot 1_{\Phi(x)<0.5} + (1 - \Phi^*(x)) 1_{\Phi(x)\geq 0.5} \\
& \quad - \Phi^*(x) \cdot 1_{\Phi^*(x)<0.5} - (1 - \Phi^*(x)) 1_{\Phi^*(x)\geq 0.5} \\
& = (2\Phi^*(x) - 1) \cdot 1_{\Phi(x)<0.5} \cdot 1_{\Phi^*(x)\geq 0.5} \\
& \quad + (1 - 2\Phi^*(x)) \cdot 1_{\Phi(x)\geq 0.5} \cdot 1_{\Phi^*(x)<0.5} \\
& \leq \left(\frac{\Phi^*(x)}{\Phi(x)} - 1\right) \cdot 1_{\Phi(x)<0.5} \cdot 1_{\Phi^*(x)\geq 0.5} \\
& \quad + \left(1 - \frac{\Phi^*(x)}{\Phi(x)}\right) \cdot 1_{\Phi(x)<0.5} \cdot 1_{\Phi^*(x)\geq 0.5} \\
& \leq \left(\frac{c_2}{c_1} - 1\right) \cdot 1_{\Phi(x)<0.5} \cdot 1_{\Phi^*(x)\geq 0.5} \\
& \quad + \left(1 - \frac{c_1(1-\epsilon)}{c_2}\right) \cdot 1_{\Phi(x)<0.5} \cdot 1_{\Phi^*(x)\geq 0.5}
\end{aligned}
$$

and

$$
\begin{aligned}
\mathcal{R}(\Phi) - \mathcal{R}(\Phi^*) & = \mathbb{E}\left[\mathcal{R}_x(\Phi) - \mathcal{R}_x(\Phi^*)\right] \\
& \leq \max\left\{\frac{c_2}{c_1} - 1, 1 - \frac{c_1(1-\epsilon)}{c_2}\right\}.
\end{aligned}
$$

# B  Experiment details

The data sets are divided into training and test sets. For VPU, a cross-validation criterion is provided, so we further proportionally divide the training set into training and validation sets.

Table 5: Description of UCI datasets used in experiments.

| Dataset | $N$ | size of test set | $d$ |
|---|---|---|---|
| Page Blocks | 3284 | 2189 | 10 |
| Grid Stability | 6000 | 4000 | 14 |
| Avila | 10430 | 10437 | 10 |

Table 6: Experimental settings for UCI datasets. $N_P, M, M_v$ denote respectively the number of positive samples in the training set, number of labeled positive samples in the training set, number of labeled positive samples in the validation set. The size of validation unlabeled samples can be calculated via $N_v = N \times M_v/M$, where $N$ is the size of training unlabeled samples.

| Experiment | setting | Data amount | Validation size | $\pi_P$ | Hyperparameter |
|---|---|---|---|---|---|
| Page Blocks[1] | '2,3,4,5' vs '1' | $N_P$=342 $M$=100 | $M_v$=16 | 0.104 | $\lambda = 0.0003, \alpha = 0.3$ |
| Page Blocks[2] | '1' vs '2,3,4,5' | $N_P$=2942 $M$=100 | $M_v$=16 | 0.896 | $\lambda = 0.0001, \alpha = 0.3$ |
| Grid Stability[1] | 'stable' vs 'unstable' | $N_P$=2187 $M$=1000 | $M_v$=167 | 0.365 | $\lambda = 0.1, \alpha = 0.3$ |
| Grid Stability[2] | 'unstable' vs 'stable' | $N_P$=3813 $M$=1000 | $M_v$=167 | 0.635 | $\lambda = 0.1, \alpha = 0.3$ |
| Avila[1] | 'A' vs The rest | $N_P$=4286 $M$=2000 | $M_v$=192 | 0.411 | $\lambda = 0.1, \alpha = 0.3$ |
| Avila[2] | 'A,F' vs The rest | $N_P$=6247 $M$=2000 | $M_v$=192 | 0.599 | $\lambda = 0.03, \alpha = 0.3$ |

For each experiment, 10 repeated runs are done, and mean and standard variance of test accuracy are calculated. By default, for each run the neural network is trained for 50 epochs, and results are reported at the epoch with lowest Kullback-Leibler divergence on the validation set. We fix $\alpha$ to 0.3 and use the Kullback-Leibler divergence on the validation set as the criterion for tuning $\lambda$, selected in $\{1e-4, 3e-4, 1e-3, \cdots, 1, 3\}$.

Moreover, we denote $\mathbb{P}(y = +1), \mathbb{P}(y = -1)$ by $\pi_P$ and $\pi_N$.

## B.1 UCI datasets

We first clarify the UCI datasets used in our experiments in Table 5. Then, we give the detailed experimental settings of each experiment in Table 6. The datasets do not go through any preprocessing.

## B.2 FashionMNIST, CIFAR-10 and STL-10

Labels of ten classes of each image datasets are reported in Table. 7, which are denoted by numbers 0 to 9 in Section 5.3. The details of the experiments are shown in Table 8. All datasets conduct data preprocessing: normalization with mean and standard deviation both as 0.5 at all dimensions.

## B.3 Choice of hyperparameters of GenPU

GenPU contains four hyperparameters: $\pi_P\lambda_p$, $\pi_P\lambda_u$, $\pi_N\lambda_n$, $\pi_N\lambda_u$. Although the parameters are coupled for given $\pi_P$ in [14], our experience shows that the better performance can be achieved by selecting the four parameters independently. Table 9 shows the best hyperparameters which lead to the largest classification accuracies on test sets. They are selected in $\{0.01, 0.05, 0.1, 0.5, \ldots, 1000, 5000\}$ by greedy grid search.

| FashionMNIST | t-shirt, trouser, pullover, dress, coat, sandal, shirt, sneaker, bag, ankle boot |
|---|---|
| CIFAR-10 | airplane, automobile, bird, cat, deer, dog, frog, horse, ship, truck |
| STL-10 | airplane, bird, car, cat, deer, dog, horse, monkey, ship, truck |

Table 7: Class labels of image datasets, which are denoted by numbers $0, 1, \ldots 9$ in Section 5.3.

Table 8: Experimental settings for FashionMNIST, CIFAR-10 and STL-10. $N_P, M, M_v$ denote respectively the number of positive samples in the training set, number of labeled positive samples in the training set, number of labeled positive samples in the validation set. The size of validation unlabeled samples can be calculated via $N_v = N \times M_v/M$, where $N$ is the size of training unlabeled samples.

| Experiment | Setting | Data amount | Validation size | $\pi_P$ | Hyperparameter |
|---|---|---|---|---|---|
| FashionMNIST[1] | '1,4,7' vs '0,2,3,5,6,8,9' | $N_P$=15000 $M$=3000 | $M_v$=500 | 0.300 | $\lambda = 0.3, \alpha = 0.3$ |
| FashionMNIST[2] | '0,2,3,5,6,8,9' vs '1,4,7' | $N_P$=39000 $M$=3000 | $M_v$=500 | 0.700 | $\lambda = 3, \alpha = 0.3$ |
| CIFAR-10[1] | '0,1,8,9' vs '2,3,4,5,6,7' | $N_P$=17000 $M$=3000 | $M_v$=500 | 0.400 | $\lambda = 0.03, \alpha = 0.3$ |
| CIFAR-10[2] | '2,3,4,5,6,7' vs '0,1,8,9' | $N_P$=27000 $M$=3000 | $M_v$=500 | 0.600 | $\lambda = 0.01, \alpha = 0.3$ |
| STL-10[1] | '0,2,3,8,9' vs '1,4,5,6,7' | $N_P$=100000 $M$=2500 | $M_v$=250 | $unknown$ | $\lambda = 0.3, \alpha = 0.3$ |
| STL-10[2] | '1,4,5,6,7' vs '0,2,3,8,9' | $N_P$=100000 $M$=2500 | $M_v$=250 | $unknown$ | $\lambda = 0.1, \alpha = 0.3$ |

Table 9: Choice of hyperparameters for GenPU.

| Dataset | $\pi_P\lambda_p$ | $\pi_P\lambda_u$ | $\pi_N\lambda_n$ | $\pi_N\lambda_u$ |
|---|---|---|---|---|
| FashionMNIST | 0.01 | 1 | 100 | 1 |
| | 0.01 | 1 | 1000 | 50 |
| CIFAR-10 | 0.01 | 1 | 100 | 1 |
| | 0.01 | 1 | 100 | 1 |
| Page Blocks | 0.01 | 1 | 1000 | 1 |
| | 0.01 | 1 | 200 | 1 |
| Grid Stability | 0.01 | 1 | 1000 | 500 |
| | 0.01 | 1 | 1000 | 500 |
| Avila | 0.01 | 1 | 100 | 1 |
| | 0.001 | 1 | 1000 | 500 |

## B.4 Comparison with known $\pi_P$

In Table 10, we compare the classification accuracies of VPU, nnPU and uPU on UCI and image datasets. All the settings are the same as in the main body of the paper, except that the true value of $\pi_P$ is assumed to be known for nnPU and uPU. Notice that the experiment on STL-10 is not performed because the exact $\pi_P$ is unavailable.

Table 10: Classification accuracies (%) of compared methods, where $*$ means that the algorithm is performed with the true value of $\pi_P$.

| Dataset | Page Blocks[1] | Page Blocks[2] | Grid Stability[1] | Grid Stability[2] | Avila[1] | Avila[2] |
|---|---|---|---|---|---|---|
| VPU | **93.6 ± 0.4** | **93.5 ± 0.7** | **92.6 ± 0.3** | 89.5 ± 0.5 | **82.0 ± 0.9** | **87.2 ± 0.5** |
| nnPU$^*$ | 92.3 ± 1.2 | 91.7 ± 0.6 | 91.5 ± 1.7 | **90.5 ± 0.3** | 75.9 ± 2.2 | 84.8 ± 0.5 |
| uPU$^*$ | 93.0 ± 1.2 | 90.0 ± 2.8 | 92.2 ± 0.1 | 87.9 ± 0.9 | 76.5 ± 1.0 | 84.0 ± 1.0 |

| Dataset | F-MNIST[1] | F-MNIST[2] | CIFAR-10[1] | CIFAR-10[2] |
|---|---|---|---|---|
| VPU | **92.7 ± 0.3** | **90.8 ± 0.6** | **89.5 ± 0.1** | **88.8 ± 0.8** |
| nnPU$^*$ | 92.1 ± 0.3 | 90.7 ± 1.4 | 87.2 ± 0.7 | 86.5 ± 1.7 |
| uPU$^*$ | 90.4 ± 1.4 | 74.1 ± 1.9 | 79.1 ± 2.4 | 68.7 ± 0.4 |

Figure 4: Positive (a) and negative (b) samples generated by GenPU on FashionMNIST with '1, 4, 7' as positive labels

Table 11: The class prior estimated by KM2 under selection bias with the true class prior $\pi_P = 0.3$

| $n_1/n_4$ | 1 | 2 | 3 | 4 | 5 | 6 | 7 | 8 | 9 | 10 |
|---|---|---|---|---|---|---|---|---|---|---|
| estimated $\pi_P$ | 0.267 | 0.249 | 0.206 | 0.188 | 0.164 | 0.170 | 0.151 | 0.157 | 0.150 | 0.144 |

## B.5 Mode collapse of GenPU

The failure of GenPU in the experiments is caused by mode collapse. This is demonstrated in Fig. 4, which shows the positive (a) and negative (b) images generated by GenPU. Positive labels ('Positive' vs 'Negative') are given by '1,4,7' (Trouser, Coat, Sneaker) vs '0,2,3,5,6,8,9' (T-shirt/Top, Pullover, Dress, Sandal, Skirt, Bag, Ankle boot). We observe that, in spite of the good quality of the generated images, some modes are neglected be the generators.

## B.6 KM2, nnPU and uPU under selection bias

Table 11 shows that the class prior estimation method KM2 significantly affected by the selection bias, which also yields poor performance of nnPU. As can be observed in Fig. 5, nnPU is even more robust to selection bias if the accurate $\pi_P$ is known a priori.

## B.7 Alternative regularization terms

Mixup is a powerful regularization technique, but it might not be applicable to domains other than image. Besides, its data-augmentation nature undermines credibility of VPU's superiority shown in the experiments. In fact, some other forms of regularization also work well, such as adversarial training [37] and virtual adversarial training [38]. Here we introduce a large-margin regularization term, proposed in [39], as an alternative for the Mixup-based regularization. It penalizes the positive instances that are misclassified by $\Phi$ or have small margins between $\log \Phi(x)$ and $\log(1 - \Phi(x))$.

Figure 5: Comparison of PU methods under selection bias of $\mathcal{P}$, with accurate class prior $\pi_P$ known for uPU and nnPU

Table 12: Classification accuracies (%) on image and UCI datasets of experiments with the same setting as in Section 5.2 and 5.3. VPU w/ Mixup is the VPU we develop in main body of this paper, while VPU w/ margin replaces the regularization with the large-margin loss (13).

| Dataset | Page Blocks[1] | Page Blocks[2] | Grid Stability[1] | Grid Stability[2] | Avila[1] | Avila[2] |
|---|---|---|---|---|---|---|
| VPU w/ Mixup | $93.6 \pm 0.4$ | $93.5 \pm 0.7$ | $\mathbf{92.6 \pm 0.3}$ | $89.5 \pm 0.5$ | $\mathbf{82.0 \pm 0.9}$ | $\mathbf{87.2 \pm 0.5}$ |
| VPU w/ margin | $\mathbf{95.6 \pm 1.3}$ | $\mathbf{94.0 \pm 0.6}$ | $\mathbf{92.6 \pm 0.3}$ | $\mathbf{90.5 \pm 0.5}$ | $81.4 \pm 0.3$ | $86.8 \pm 0.5$ |
| nnPU | $93.4 \pm 1.1$ | $90.2 \pm 2.6$ | $80.8 \pm 2.5$ | $84.1 \pm 1.8$ | $73.3 \pm 2.0$ | $83.1 \pm 2.1$ |
| Dataset | F-MNIST[1] | F-MNIST[2] | CIFAR-10[1] | CIFAR-10[2] | STL-10[1] | STL-10[2] |
| VPU w/ Mixup | $\mathbf{92.7 \pm 0.3}$ | $90.8 \pm 0.6$ | $\mathbf{89.5 \pm 0.1}$ | $88.8 \pm 0.8$ | $\mathbf{79.7 \pm 1.5}$ | $\mathbf{83.7 \pm 0.1}$ |
| VPU w/ margin | $92.6 \pm 0.4$ | $\mathbf{91.1 \pm 0.2}$ | $89.2 \pm 0.2$ | $\mathbf{88.9 \pm 0.3}$ | $74.5 \pm 0.9$ | $82.6 \pm 1.5$ |
| nnPU | $90.8 \pm 0.6$ | $90.5 \pm 0.4$ | $85.6 \pm 2.3$ | $85.5 \pm 2.0$ | $78.3 \pm 1.2$ | $82.2 \pm 0.5$ |

It is a smooth version of $\max\left\{0, \log\left(1 - \Phi\left(x\right)\right) + \log\alpha - \log\Phi\left(x\right)\right\}$ and formulates as

$$\mathcal{L}_{reg-margin}\left(\Phi\right) = \text{softplus}\left(\log\left(1 - \Phi\left(x\right)\right) + \log\alpha - \log\Phi\left(x\right)\right)$$
$$= \log\left(1 + \alpha\frac{1 - \Phi\left(x\right)}{\Phi\left(x\right)}\right). \tag{13}$$

Table 12 reports the results of experiments with the same setting as in Section 5.2 and 5.3. Though not as good as the Mixup-based regularization, the large-margin regularization significantly outperforms nnPU in most experiments.

## B.8 Other metric for comparison

Accuracy might not be the best metric, especially when data sets are imbalanced. Therefore, except accuracy shown in the main body, we here also report in Table 13 the area under curve (AUC) values of experiments on image datasets

## B.9 nnPU with Mixup

To further demonstrate the advantage of VPU over nnPU, we also conduct experiments on nnPU on FashionMNIST with unlabeled data augmented by MixUp. The classification accuraries are reportd in Table 14, which shows that nnPU does not significantly benefit from Mixup.

Table 13: AUC values of compared methods on FashionMNIST (abbreviated as "F-MNIST"), CIFAR-10 and STL-10 datasets. Experiment settings are the same as in Section 5.3.

| Dataset | F-MNIST[1] | F-MNIST[2] | CIFAR-10[1] | CIFAR-10[2] | STL-10[1] | STL-10[2] |
|---------|-----------|-----------|-------------|-------------|-----------|-----------|
| VPU | **0.973 ± 0.002** | **0.957 ± 0.005** | **0.956 ± 0.001** | **0.953 ± 0.003** | **0.976 ± 0.002** | **0.963 ± 0.003** |
| nnPU | 0.961 ± 0.004 | 0.945 ± 0.005 | 0.954 ± 0.003 | 0.953 ± 0.002 | 0.850 ± 0.007 | 0.898 ± 0.004 |
| uPU | 0.955 ± 0.006 | 0.918 ± 0.008 | 0.952 ± 0.003 | 0.949 ± 0.004 | 0.823 ± 0.013 | 0.862 ± 0.014 |
| Genpu | 0.673 ± 0.018 | 0.868 ± 0.007 | 0.790 ± 0.012 | 0.811 ± 0.014 | 0.789 ± 0.004 | 0.793 ± 0.011 |
| RP | **0.973 ± 0.001** | 0.954 ± 0.002 | 0.953 ± 0.002 | 0.951 ± 0.003 | 0.829 ± 0.019 | 0.851 ± 0.015 |

Table 14: Classification accuracies (%) of nnPU with Mixup on FashionMNIST. Experiment settings are the same as in Section 5.3. The * mark indicates accurate class prior known.

| | VPU | nnPU | nnPU+MixUp | nnPU* | nnPU*+MixUp |
|---|-----|------|------------|-------|-------------|
| F-MNIST[1] | **92.7 ± 0.3** | 90.8 ± 0.6 | 91.0 ± 0.6 | 92.1 ± 0.3 | 92.4 ± 0.5 |
| F-MNIST[2] | **90.8 ± 0.6** | 90.5 ± 0.4 | 89.9 ± 0.3 | 90.7 ± 1.4 | 90.7 ± 0.4 |

## C  Extension

One alternative to the variational loss is

$$
\begin{aligned}
\mathcal{L}_{\mathrm{JS}}(\Phi) &= \max_{D:\mathbb{R}^d \mapsto [0,1]} \mathbb{E}_{f_P}\left[\log D(x)\right] + \mathbb{E}_{f_\Phi}\left[\log\left(1 - D(x)\right)\right], \\
&= \max_{D:\mathbb{R}^d \mapsto [0,1]} \int f_P(x) \log D(x) + f_\Phi(x) \log\left(1 - D(x)\right) \mathrm{d}x.
\end{aligned}
$$

Here $D$ can be interpreted as a discriminator as in GAN, which intends to separate the samples drawn from $f_P$ and those obtained by sampling from $f_\Phi$. By setting

$$
\frac{\partial \left(f_P(x) \log D(x) + f_\Phi(x) \log\left(1 - D(x)\right)\right)}{\partial D(x)} = 0,
$$

we can obtain that the optimal $D$ is

$$
D(x) = \frac{f_P(x)}{f_P(x) + f_\Phi(x)},
$$

and

$$
\begin{aligned}
\mathcal{L}_{\mathrm{JS}}(\Phi) &= \int f_P(x) \log \frac{f_P(x)}{\frac{1}{2}\left(f_P(x) + f_\Phi(x)\right)} \mathrm{d}x + \log \frac{1}{2} \\
&\quad + \int f_\Phi(x) \log \frac{f_\Phi(x)}{\frac{1}{2}\left(f_P(x) + f_\Phi(x)\right)} \mathrm{d}x + \log \frac{1}{2} \\
&= 2\mathrm{JS}\left(f_P || f_\Phi\right) - \log 4,
\end{aligned}
$$

where $\mathrm{JS}\left(f_P || f_\Phi\right)$ denotes the Jensen-Shannon divergence between $f_P$ and $f_\Phi$. Thus, $\mathcal{L}_{\mathrm{JS}}(\Phi) - \mathcal{L}_{\mathrm{JS}}(\Phi^*) \geq 0$ for all $\Phi$ since $f_P = f_{\Phi^*}$. In practice, we can approximate $D$ by another neural network, and minimize $\mathcal{L}_{\mathrm{JS}}$ by adversarial learning.

Another choice of variational loss can be derived from a weighted $L^2$ distance between $f_\Phi$ and $f_P$ as

$$
\begin{aligned}
\int f(x)^{-1} \left( f_\Phi(x) - f_P(x) \right)^2 \mathrm{d}x &= \frac{\int f(x)\Phi(x)^2 \mathrm{d}x}{\mathbb{E}_f[\Phi(x)]^2} - 2\frac{\int f_P(x)\Phi(x)\mathrm{d}x}{\mathbb{E}_f[\Phi(x)]} \\
&\quad + \int f(x)^{-1} f_P(x)^2 \mathrm{d}x, \\
&= \frac{\mathbb{E}_f[\Phi(x)^2]}{\mathbb{E}_f[\Phi(x)]^2} - 2\frac{\mathbb{E}_{f_P}[\Phi(x)]}{\mathbb{E}_f[\Phi(x)]} \\
&\quad + \int f(x)^{-1} f_P(x)^2 \mathrm{d}x, \\
&= \mathcal{L}_2(\Phi) + \int f(x)^{-1} f_P(x)^2 \mathrm{d}x,
\end{aligned}
$$

where

$$
\mathcal{L}_2(\Phi) \triangleq \frac{\mathbb{E}_f[\Phi(x)^2]}{\mathbb{E}_f[\Phi(x)]^2} - 2\frac{\mathbb{E}_{f_P}[\Phi(x)]}{\mathbb{E}_f[\Phi(x)]}
$$

and $\int f(x)^{-1} f_P(x)^2 \mathrm{d}x$ is a constant independent of $\Phi$. It can be seen from the above that the loss $\mathcal{L}_2$ satisfies

$$
\begin{aligned}
\mathcal{L}_2(\Phi) - \mathcal{L}_2(\Phi^*) &= \int f(x)^{-1} \left( f_\Phi(x) - f_P(x) \right)^2 \mathrm{d}x \\
&\geq 0.
\end{aligned}
$$