[Reviews · NeurIPS 2020]

Review 1

Summary and Contributions: This paper focuses on the binary classification positive-unlabeled (PU) learning setting where only positive and unlabeled data is available, and proposes a new variational approach that has the advantages of (A) not requiring explicit computation of the class prior and (B) not requiring assumptions of separability for theoretical guarantees. The proposed approach also leverages the MixUp regularization strategy, and show superior performance compared to a range of recent and standard PU learning approaches on standard benchmarks and datasets.

Strengths: - (S1) The proposed approach has the benefit of not requiring explicit computation of a class prior, and as the authors point out, while it still requires selection of hyperparameters on a (PU) labeled validation dataset, this is preferable to trying to compute the class prior on a PU labeled dataset - (S2) The proposed approach shows very strong results compared to a range of recent approaches on a range of datasets and tasks - (S3) The paper shows necessary ablations and micro-studies, for example on the effect of regularization strategy, labeled data amount, and positive labeling data bias - (S4) The paper shows straight forward but clean theoretical guarantees given the simple input assumptions used

Weaknesses: - (W1) The proposed approach still relies on the "selected completely at random" (SCAR) assumption; this is a common but strong one used in the PU learning literature (however, the authors do show that empirically their approach does seem to perform well without this assumption, i.e. w selection bias w.r.t. classes in underlying non-binary datasets, which is positive) - (W2) The proposed approach crucially relies on an 'irreducibility' assumption (Assumption 2) which seems innocent enough, but from which the main theoretical results follow quite directly... given this importance (which the authors acknowledge), a more extensive and helpful exposition around it would have improved the paper greatly

Correctness: To this author's knowledge, yes.

Clarity: The paper is very clearly structured and written.

Relation to Prior Work: Yes, prior work is clearly outlined and discussed, including a helpful comparison table upfront.

Reproducibility: Yes

Additional Feedback: See (W2) above. *Notes in response to author rebuttal:* - I appreciate the additional analysis on Assumption #2 - I noted in original review that I believe your treatment of SCAR assumption with the empirical non-SCAR study was good; just noting that this is still a strong assumption as far as the theory goes... - Overall, my review remains unchanged!


Review 2

Summary and Contributions: This paper introduces VPU to learn an ideal PU classifier by utilizing the upper bound of variational loss. VPU avoids approximating the class prior, which authors claim to be an important hurdle in the PU learning.

Strengths: This paper clearly enumerates the used assumptions and the potential characteristics of the classifier inference procedure through a number of theorems. Particularly, Theorem 7 provides an important bound in the classifier risk while its assumptions can be problematic in some contexts (see the below comments)

Weaknesses: 1. The trustworthiness of f_p In spite of the clear assumption statements, I have concerns of utilizing f_p in the given setting. I am comfortable up to the derivation of Theorem 6 and Eq 6. However, the authors use Eq 7 to optimize the KL divergence, and Eq 7 uses the expectation with the distribution of f_p. While the paper asserts that the distribution function, f_p, can be approximated by the positive dataset. However, Algorithm 1 uses the sample minibatch of B^P to empirically estimate f_p. To my understanding, the size of B^P would be limited because it is "mini-batch" as the author mentioned. 2. Usefulness/Practical meaning of Assumption 2 Being stated as an assumption, Assumption 2 can be set as authors or authors of [22]. However, its practicality needs to be assessed from a different perspective. I can agree that there will be a set A to show \Phi*(x)=1 of every x in A. However, such A can be a very small set to include a trivial case. From the practical perspective, the size of A and its convergence of A to A* when \integral_{A*}f_p(x)dx=1. In plain words, A needs to cover the complete positive set, eventually. In this sense, there should be further investigation on either A's expansion or \Phi*'s error bound in conjunction to A's size. 3. Use of Mix-Up Actually, this is a follow-up of Question 1. Since f_p is not completely recovered or approximated, authors are using data-augmentation to supplement the incompleteness of f_p estimation used for the calculation of E_f_p[log\Phi(x)]. This can be seen that authors created the f_p estimation problem and covered the problem by the data-augmentation. 4. Assumption on Theorem 7 Authors proved Theorem 7 by assuming M,N goes to infinity. However, the PU learning is built upon an assumption that we have a limited Positively labeled dataset and a huge Unlabeled dataset. From this aspect of label imbalance, Theorem 7 looses its significance. Having said that, I appreciated the authors' trial on providing the risk bound.

Correctness: It seems that the authors covered most of important baselines with diverse datasets. I went through the formula derivations, and it seems to be fine.

Clarity: Authors stated "... subject to constraints \Phi(x) \in [0,1] and max_x \Phi(x)=1." on Line 94, page 3. Do you need "max_x \Phi(x)=1" when "\Phi(x) \in [0,1]" is already stated?

Relation to Prior Work: Prior works are clearly stated and discussed.

Reproducibility: Yes

Additional Feedback: Please respond to the weakness part of questions with focus on Question 1 and 2. //////////////////////// Added after reading the rebuttal The author response does not resolve my concerns, so unfortunately, I have to maintain my position on the rejection. The authors claim that they do not reply on the class prior, but actually, they do. If you look at Eq 6, you can find E_f_P[log\Phi(x)]. Here, f_P is the class prior that they estimated. Then, the question is how they estimated f_P, and they estimate f_P by the samples of the mini-batch, and even they admitted f_P has high variance because of the small samples. Additionally, they make an implicit assumption that the mini-batch must contain the positive instances by the class proportion, so they rely on SCAR. Given these pieces of information, I conclude that they are just using an empirical class prior under the SCAR assumption. This is no removal of class prior. The authors acknowledge the limitation of Theorem 7 by stating that they are working on the M<<N case. Truly, if you just assume M and N goes infinity, this goes against the basic foundation of the PU learning, which is learning under the limited positively labeled instances with massive unlabeled instances. By the above arguments, I see that I have troubles in providing consensus to the acceptance, and I have to argue for the rejection. Thanks


Review 3

Summary and Contributions: This paper proposes a binary classification method that can learn from positive and unlabeled data, instead of the traditional setup where both positive and negative data are available. This problem setting is known as positive-unlabeled (PU) learning, but previous works had to rely on a preparation step of estimating the class prior before learning the classifier with the PU method. On the other hand, this paper propose a variational principle for PU learning, which does not involve class prior estimation or other intermediate estimation steps, by only introducing an assumption similar to "irreducibility" used in mixture proportion estimation literature. The proposed method suffers from overfitting and the paper further extend the algorithm to include a mixup step. It gives theoretical analysis such as the asymptotic correctness. Experiments show the proposed method works better than previous methods.

Strengths: - The novelty is high: the paper proposes a new variational approach for PU learning. - The method makes PU learning simpler: it removes the intermediate step of class prior estimation. - It works better than previous methods in experiments. - Theoretical insights are given in the paper on asymptotic correctness and for relaxation of some of the main assumptions used for deriving the method. - The organization and story is good and the motivation is clear.

Weaknesses: The proposed method utilizes mixup to alleviate overfitting issues. Mixup is a powerful data augmentation technique, so it makes it harder to observe why the proposed method works better than previous methods. My suggestion is to compare without the mixup component. This is partially reported in the ablation studies for Fashion-MNIST, and it seems that the proposed method will perform worse than previous methods without this regularization technique. Comparing with extensions of previous methods that incorporate mixup will also be meaningful to cancel out the mixup component. In Table 10, VPU performs better even when the true class prior is given to previous methods. Does this mean the advantage of VPU's high test accuracy is not coming from the removal of intermediate step of class prior estimation? (Maybe related to previous comment.) For the above reasons, I found it difficult to gain a good understanding of the proposed method through the experiments.

Correctness: Yes.

Clarity: Yes, the paper is written well.

Relation to Prior Work: The relationship to prior work is explained carefully in the paper. The main contribution and difference is that this paper does not require the intermediate class prior estimation step.

Reproducibility: Yes

Additional Feedback: Minor comments: - in appendix line 381: underestmated --> underestimated - Using mixup may prohibit the proposed algorithm to be used in domains other than images. ------------------------------------------- ------------------------------------------- Thank you for answering my questions. I appreciate the additional experiments with a fair baseline (nnPU + Mixup), and it looks like the proposed method still works better compared with nnPU + Mixup. If accepted, I would like to suggest to include the updated baseline for the other datasets too. My other concern was about the story of the paper. I think the motivation was based on the fact that previous methods suffer from inaccurate estimates coming from class prior estimation, and the proposed method VPU doesn't rely on class prior which leads to better generalization. However in Table 10 in Appendix, VPU performs better even when the true class prior is given to previous methods (except for one case out of 10). This implies that the advantage of VPU's high test accuracy is not coming from the removal of class prior estimation, and somehow seems to contradict with the story of the paper. Since this part is related to the main message of the paper, I think further empirical investigation of this point is important.


Review 4

Summary and Contributions: The paper introduces a principled method for positive-unlabeled learning (small set of positive labeled examples, large set of unlabeled examples) that overcomes shortcomings of prior approaches in that it both does not depend on positive-negative class separability for consistency, and does not require estimating the prior probability of the positive class. They use a variational bound on the divergence between the true and estimated positive data distribution and show a number of useful theoretical results including consistency given some pretty reasonable assumptions. They also introduce an interpolation-smoothness regularization and demonstrate its effectiveness. They thoroughly evaluate the proposed method along different dimensions with a variety of experiments and data sets - showing improvement compared to prior state-of-the-art approaches as well as demonstrating robustness to sample selection bias and number of positive samples and positive labeled samples. The paper is very clearly written and well-organized, and very thorough.

Strengths: -The idea of using the variational bound is interesting and novel for PU (to my knowledge) - with the shortcomings it addresses from previous work clearly pointed out. The assumptions also seem very reasonable. The method makes a lot of sense and is well-validated theoretically and experimentally. -The work is very relevant - as PU is a growing and useful problem of interest to the community (as many real world and modern use case problems could fit this scenario) - and I found the work inspirational and possibly useful for motivating ideas in other areas as well. -The idea of the regularization used is also interesting. -The paper is very well-written and organized, and very clear and easy to understand. Every part from description of related work, presentation and explanation of theoretical results, method and algorithm, and experimental study are all very clear and thorough. -The experimental and ablation study is very thorough and provides detailed analyses of the proposed methods and comparison with other approaches.

Weaknesses: -A key limitation is the use of only the accuracy metric. Looking at the data sizes and splits in the supplement, the data sets used are at least somewhat imbalanced. As such accuracy may not be the best metric to use for comparing different classifiers. Ideally additional metrics would also be reported, including AUC and different rank and correlation based metrics (e.g., Matthews Correlation Coefficient, Gini score0. However for many cases the data does not seem to be too imbalanced - so it's not a show-stopper. -Variational bounds and minimizing KL divergence is not new, although it seems to be for this problem - and the final loss term ends up being quite similar to the typical sort of losses used for this type of problem - so it is not too much of novelty in terms of the loss itself - but there is additional supporting theory, experiments, and regularization (which is also adapted). -There are a few unclear points and questions raised: --In the algorithm - how do you prevent the log terms from taking negative infinity values causing the training to diverge, as phi can output a value of 0? --In the last line of the algorithm, it shows normalizing the output of phi by dividing by the max value of phi 0 why is this necessary? What does this have to do with Assumption 2? This is not clear, and not explained. --Why is alpha chosen as 0.3? Any hyper-parameter sensitivity analysis to show the impact of different alpha?

Correctness: Both seemed correct to me.

Clarity: The paper was very well written - the most well written, organized, and thorough of the papers I was assigned to review

Relation to Prior Work: Related work is clearly discussed and contrasted.

Reproducibility: Yes

Additional Feedback: (see comments and questions above) ***Update after discussion and author response*** I found the author response to be helpful. I think most negative points brought by myself and the other reviewers were pretty minor and largely addressed by the authors' response comments. I do not agree that the proposed method relies on estimating the class prior like past methods - i.e., f_p is the positive data distribution - the class prior needed to be provided (or estimated) in past methods is the P(y=+1). It's much more reasonable to assume a decent idea of the positive data distribution could be had from the positive unlabeled samples (as opposed to a completely unknown proportion of positive and negatives in the data distribution needed to estimate the class prior), and as they show, even under sample selection bias this bound minimization approach still seems to work well. Maybe more analyses is needed in the future but the results are of interest to the community. Therefore I'm keeping my rating, I found the approach to be interesting, novel and useful/effective, with some fairly reasonable and clear theoretical justification as well.

[Author Response · NeurIPS 2020]

We would like to thank the reviewers for their insightful feedback. In the following, we address their key concerns.

**R1, R2, R4: Suggest more extensive analysis on Assumption 2 and the normalization step in Algorithm 1.**

Following reviewers' suggestions, we will add more thorough analysis in the final paper.

1. A brief explanation of Assumption 2: a) There is an anchor set $\mathcal{A}$ so that $\mathbb{P}(y = +1|x \in \mathcal{A}) = 1$, where $\mathcal{A}$ has a
positive probability under $f_P$. This is a strong variant of the widely used irreducibility assumption [22] (Sec. A.2 in
Suppl. Material). b) As pointed out by **R2**, $\mathcal{A}$ could be a small set. But there are *almost surely* samples in $\mathcal{A}$ as data
size tends to infinity according to the assumption even if $\mathcal{A}$ does not cover the whole positive data set $\mathcal{P}$. c) In practical
cases where $\mathcal{A}$ is too small and $|\mathcal{P}|$ is finite, $\mathcal{A} \cap \mathcal{P}$ could be empty. So we analyzed the misclassification rate under a
relaxation of the assumption (Condition (ii) of Assumption 6 and Theorem 7).

2. The normalization step comes from Assumption 2 as indicated by **R4**. Without the step, the variational loss $\mathcal{L}_{\text{var}}$
can be minimized by $\Phi = c \cdot \Phi^*$ for all $c > 0$ (Remark 4) due to the fact that the Bayesian classifier $\Phi^* \propto f_p/f$ is
only identifiable up to a multiplicative constant without Assumption 2.

**R2, R3: VPU seems to heavily rely on Mixup. Its advantages and applications are then limited.**

Mixup was introduced in VPU as a regularizer to solve the overfitting problem (Table 4 and Lines 100–105, 376–384).
We will conduct extensive comparison experiments for analysis of Mixup. A part of the results (percent accuracy) on
the FashionMNIST (FM) dataset is shown in the table below, where class priors are estimated by KM2 [28] for nnPU.

17
|  | VPU (Mixup) | nnPU (Mixup) | VPU (Large-margin) |  | VPU (Mixup) | nnPU (Mixup) | VPU (Large-margin) |
|---|---|---|---|---|---|---|---|
| FM[1] | $92.7 \pm 0.3$ | $91.0 \pm 0.6$ | $92.6 \pm 0.4$ | FM[2] | $90.8 \pm 0.6$ | $90.5 \pm 0.7$ | $91.1 \pm 0.2$ |

1. In many advanced methods (e.g., nnPU), the class prior is considered as a given constant, and it is difficult for
Mixup to reduce the error caused by the inaccurate class prior estimation. Specifically, nnPU already has an effective
strategy to tackle overfitting, and its accuracy is not significantly improved by Mixup as shown in the table.

2. In the case mentioned by **R3**, where Mixup is not applicable, the overfitting of VPU can be solved by regularization
techniques without data augmentation (e.g., large-margin regularization). This will be investigated in the final paper,
and the feasibility can be partially demonstrated by columns 4 and 8 in the above table.

3. As indicated by **R2**, a small mini-batch $\mathcal{B}^{\mathcal{P}}$ yields a unbiased but high-variance estimate of $\mathbb{E}_{x \in \mathcal{P}}[\log \Phi(x)]$. The
Mixup is necessary only if $f_P$ cannot accurately characterized by $\mathcal{P}$; otherwise the variance can be reduced by simply
scaling up $\mathcal{B}^{\mathcal{P}}$ since labeled and unlabeled mini-batches are independently drawn with the same size in Algorithm 1.

**R1: The proposed approach still relies on the "selected completely at random" (SCAR) assumption.**

For this problem, we demonstrated the performance of VPU by experiments without SCAR (Fig. 2), and performed
the asymptotic analysis by assuming the selection bias is bounded (Condition (i) of Assumption 6 and Theorem 7).

**R2: Theorem 7 loses its significance under the assumption that $M, N \to \infty$.**

We are now working on the error analysis for VPU with finite samples similar to that for nnPU [12], but this is beyond
the scope of this paper. In addition, our experiments indicated that VPU can achieve high accuracies with $M \ll N$
(Table 8), and the ablation study on data size was shown in Fig. 3.

**R3: In Table 10, VPU performs better even when the true class prior is given to previous methods.**

In fact, nnPU slightly outperforms VPU on a learning task of "Grid Stability" as shown in Table 10. Although
the comparison between VPU and uPU/nnPU with exact class priors requires further investigation, our experimental
experience shows that the Mixup does not play an indispensable role (see our response to the second comment).

**R4: The idea of variational bounds and minimizing KL divergence is not new.**

This comment is quite enlightening. Motivated by it, we found similar variational principles developed in other fields
(e.g., Donsker-Varadhan representation of KL-divergence). We will cite the related references, and clearly state our
contributions on problem formulation, asymptotic analysis and regularization for the variational PU learning.

**R4: Only the accuracy metric is used, which is not the best when data sets are imbalanced.**

The accuracy metric was used because it is popular in the literature of PU learning [12, 14] and also due to page limit.
We will provide AUC of classifiers in the final paper.

**R4: How to prevent the log terms from taking negative infinite values? Why is alpha chosen as $0.3$?**

1. In experiments, we modeled $\log \Phi$ instead of $\Phi$ by the NN, and estimated $\log \mathbb{E}_f[\Phi]$ by the log-sum-exp function.
2. The performance is not sensitive to $\alpha \in [0.1, 0.4]$ for our experiments, which is similar to the conclusion in [23].

[Meta-Review · NeurIPS 2020]

This paper presents an improved method for learning binary classifiers from positive and unlabeled data. Prior work has required the specification of the proportion of positive data in the unlabeled data set. This parameter is difficult to estimate and the resulting classifier is sensitive to it. This paper addresses that problem by minimizing the divergence between the classifier and an ideal Bayesian classifier using variational inference. While this paper is not the first to attempt to do away with the class prior estimation problem, this paper reports better empirical performance with theoretical results on consistency. As noted by all of the reviewers, the paper is very clearly written and helpfully provides a summary table comparing and contrasting prior work with the current work. The reviewers noted that positive and unlabeled data problems are growing in prevalence and the topic is timely. Some reviewers noted the novelty of the MixUp regularization approach, though there were some concerns about the generality of this approach beyond image data. There were other concerns about the applicability of the method in that in many problems it may be reasonable to assume the class prior probability is well-known and if so other methods may perform better. And, there were some concerns about the assumption the labeled and unlabeled data are selected completely at random---though this assumption is common in the area. Overall, these weaknesses were deemed to be minor and more or less outweighed by the strengths. The discussion focused attention on the balance of strengths and weaknesses and some comments were thought to be well addressed by the authors. Overall, the novelty of the approach combined with the theoretical and experimental evidence of efficacy lead me to recommend acceptance.